# AI-based approach for transcribing and classifying unstructured emergency call data: A methodological proposal

**Dalton Breno Costa**[1], **Felipe Coelho de Abreu Pinna**[2], **Anjni Patel Joiner**[3,4], **Brian Rice**[5], **João Vítor Perez de Souza**[3,4], **Júlia Loverde Gabella**[6], **Luciano Andrade**[6], **João Ricardo Nickenig Vissoci**[3,4]*, **João Carlos Néto**[2]

1 Department of Psychology, Pontifical Catholic University of Rio Grande do Sul, Rio Grande do Sul, Brazil, 2 Department of Computer and Digital Systems Engineering, Polytechnic School, University of São Paulo, São Paulo, São Paulo, Brazil, 3 Department of Emergency Medicine, Duke University School of Medicine, Durham, North Carolina, United States of America, 4 Global Emergency Medicine Innovation and Implementation Research Center, Duke Global Health Institute, Duke University, Durham, North Carolina, United States of America, 5 Department of Emergency Medicine, Stanford University, Palo Alto, California, United States of America, 6 Department of Medicine, State University of Maringá, Marringá, Paraná, Brazil

* jnv4@duke.edu

**Data Availability Statement:** Data contains sensitive patient information and cannot be publicly shared as per the rules of the Serviço de

## Abstract

Emergency care-sensitive conditions (ECSCs) require rapid identification and treatment and are responsible for over half of all deaths worldwide. Prehospital emergency care (PEC) can provide rapid treatment and access to definitive care for many ECSCs and can reduce mortality in several different settings. The objective of this study is to propose a method for using artificial intelligence (AI) and machine learning (ML) to transcribe audio, extract, and classify unstructured emergency call data in the Serviço de Atendimento Móvel de Urgência (SAMU) system in southern Brazil. The study used all "1-9-2" calls received in 2019 by the SAMU Novo Norte Emergency Regulation Center (ERC) call center in Maringá, in the Brazilian state of Paraná. The calls were processed through a pipeline using machine learning algorithms, including Automatic Speech Recognition (ASR) models for transcription of audio calls in Portuguese, and a Natural Language Understanding (NLU) classification model. The pipeline was trained and validated using a dataset of labeled calls, which were manually classified by medical students using LabelStudio. The results showed that the AI model was able to accurately transcribe the audio with a Word Error Rate of 42.12% using Wav2Vec 2.0 for ASR transcription of audio calls in Portuguese. Additionally, the NLU classification model had an accuracy of 73.9% in classifying the calls into different categories in a validation subset. The study found that using AI to categorize emergency calls in low- and middle-income countries is largely unexplored, and the applicability of conventional open-source ML models trained on English language datasets is unclear for non-English speaking countries. The study concludes that AI can be used to transcribe audio and extract and classify unstructured emergency call data in an emergency system in southern Brazil as an initial step towards developing a decision-making support tool.

Atendimento Móvel de Urgência (SAMU). Inquiries to access data can be forwarded to the Maringá Health Department at saude@maringa.pr.gov.br or by accessing the website: http://www.maringa.pr.gov.br/saude/?cod=contato.

**Funding:** This study was supported by the AI pilot grant from the Duke Global Health Institute (to JRNV). The funders had no role in study design, data collection and analysis, decision to publish, or preparation of the manuscript.

**Competing interests:** The authors have declared that no competing interests exist.

## Author summary

In our study, we utilized artificial intelligence (AI) and machine learning (ML) to help process emergency call data in the Serviço de Atendimento Móvel de Urgência (SAMU) system in southern Brazil. These calls, often concerning emergency care-sensitive conditions (ECSCs), require quick identification and treatment. Our aim was to transcribe, extract, and categorize unstructured call data to improve response times and, ultimately, patient outcomes. Leveraging AI, we transcribed calls using Automatic Speech Recognition and then categorized them using a Natural Language Understanding model. The AI was successful in accurately transcribing the audio and categorizing the calls with high accuracy. However, we also discovered that using AI in this manner in non-English speaking countries is largely uncharted territory. Our findings suggest that AI could be a powerful tool in improving emergency care responses, especially in low- and middle-income countries, and this is an exciting step towards creating decision-support tools for these critical situations.

## Introduction

Emergency care-sensitive conditions (ECSCs), conditions which require rapid identification and treatment, are attributable to over half of all deaths worldwide [1]. Prehospital emergency care (PEC) can provide rapid treatment and access to definitive care for many ECSCs and can reduce mortality in several different settings [2]. Prior to dispatching prehospital resources, emergency calls are typically routed through emergency call centers, which then go through varying levels of standardized or unstandardized questions in order to determine the most appropriate response. Accurately identifying the nature of the emergency and dispatching a correct response are key to optimal resource management and ensuring rapid triage and treatment of ECSCs.

The *Serviço de Atendimento Móvel de Urgência* (SAMU) is the national prehospital system in Brazil. Emergency calls are routed to centralized Emergency Regulation Centers (ERC) using a national toll-free hot-line which individuals access using a single phone number "1-9-2" [3]. Currently, the responsibility of triaging and managing calls primarily falls on nurses or physicians. Disadvantages of this labor-intensive process include the possibility of human-induced errors [4] and high financial costs related to employing physicians in an emergency call center. The solution in many high-income settings has been the development of structured call-taking systems to provide a standardized process to gather information and generate a specific nature code for the call complaint, and assign a priority level for ambulance response [5]. Unfortunately, the development of these types of systems is costly and time-consuming. Prior solutions to address resource scarcity through appropriate triage of emergency calls have been proposed through multi-criteria decision analysis [4].

The application of artificial intelligence (AI) and machine learning (ML) methods are being increasingly considered to augment decision-making in many different medical fields. These tools are only recently gaining traction for use in the prehospital setting [6, 7]. While the use of AI to categorize emergency calls in low- and middle-income countries (LMICs) remains largely unexplored, the automation of call-taking and triage as a decision-making support tool can augment human-dependent triaging processes and the classifications of calls and deploy limited resources to emergency calls. In this paper, we a provide a methodological proposal to apply artificial intelligence to transcribe audio, extract and classify unstructured emergency

call data in an EMS system in southern Brazil as an initial step towards developing a decision-making support tool.

## Methods

### Study settings

All "1-9-2" calls received in 2019 by the SAMU Novo Norte ERC call center in Maringá, in the Brazilian state of Paraná in 2019 were included in the analysis. Emergency calls are made to a central number by either a person having an emergency or a bystander. Calls to the ERC are answered by nursing technicians who screen and classify calls before transferring them to a physician. The physician then has the option to provide telephone guidance for lower acuity emergencies or dispatch an ambulance crew to the caller. Ambulance choices include either a basic life support unit (with driver and a nurse technician) or an advanced life support unit (with driver, a physician and a registered nurse) [8].

### Study design

Our study involves retrospective analysis of audio calls. The framework developed took an input of audio recordings and output of structured text data with relevant semantic annotations (Fig 1). The overall structure of this framework involves manual data labeling and classification, automatic speech recognition (ASR) model for Portuguese speech to text transcription, and natural language understanding (NLU) model trained to classify emergency calls and produce semantically labeled text.

### Manual screening, classification, transcription and label creation

A random subset of emergency calls was manually processed by trained research assistants (medical students fluent in Portuguese). Processing involved four steps: screening, classification, transcription and annotation. Calls were recorded as Waveform Audio File Format at a 8 khz sampling rate. Audio files included no additional information except the date and time of the call. The data was stored on Duke University's secure encrypted servers.

In the screening step, all calls were determined to be emergency calls (someone requesting emergency medical care), internal communications (calls between professionals in the SAMU 192 system), ambulance cancellations, mute (silent), ambulance delay notifications, duplicate files, prank calls, or others (including request for information and wrong number). Only emergency calls were included in subsequent analysis. Emergency calls were then classified by nature of emergency or chief complaint using a publicly available version of the categories defined in the Medical Priority Dispatch System (MPDS) schema [9]. MPDS is the most widely-used EMS dispatch system used in English-speaking high-income settings [10]. A

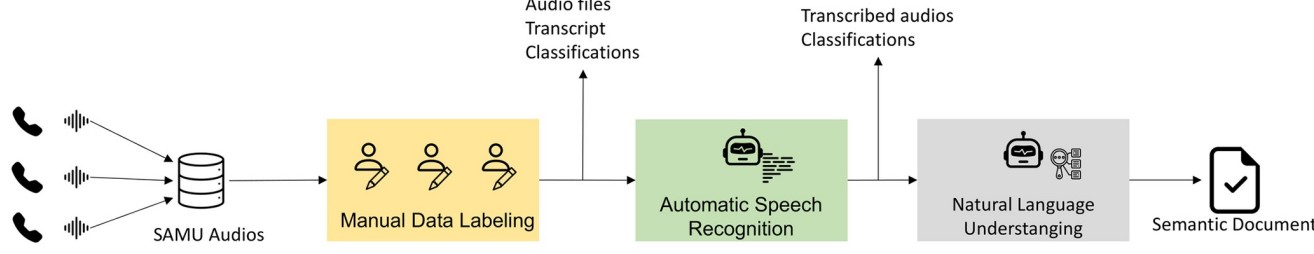

**Fig 1. Proposed methodology overview diagram.**

**Fig 2. Annotation of recorded call.**

predefined subset of the classified calls were then manually transcribed from spoken to written Portuguese using Label Studio (http://labelstud.io) tool. Finally all of the transcribed emergency calls were annotated in Label Studio to prepare them for use in an NLU model. Annotations (pictured in Fig 2) were defined by analysis of labeled calls and the final schema included the following: symptom, symptom context, body part, gestation time, animal bite, introduction, connectors, attendant speech, request, request complement, and symptom sentence.

## ASR model for Brazilian Portuguese

Automatic Speech Recognition (ASR) technology can convert audio into written text using an algorithm implemented as a computer program [11]. Current ASR systems depend on machine learning and deep learning techniques [12]. Our methods tested three open-source ASR deep learning fine-tuning pretrained models: Wav2Vec 2.0, HuBert, and WavLM [13–15]. Notably, these models are trained on large publicly available English language datasets and there is a lack of similar datasets in Brazilian Portuguese [16]. These existing models were fine-tuned to work with Brazilian Portuguese emergency calls by using the original hyperparameters from the pretrained models. The word error rate (WER) for each model was calculated using manual transcription as ground truth. The text output of the highest performing model (Wav2Vec 2.0) became the input for the subsequent step of NLU classification.

## Artificial training corpus building

For our study, the amount of labeled textual data available was insufficient to train robust neural network models. To mitigate this problem, we used an artificial example generation approach that can produce a corpus of natural language sentences from the desired domain with classification labels. Our methods employed a Backus-Naur form (BNF) grammar similar to that used in prior publications [17–19]. This grammar was constructed by the annotations made on the real transcriptions, containing regional vocabulary without the addition of supplementary vocabulary. It contains the sentence structure and vocabulary of the analyzed transcriptions, encoded in a set of derivation rules. By randomly choosing paths in expanding these rules, the grammar is capable of producing synthetic sentences similar to the real data. The examples produced contain informality, slang words, and regionalisms of the same nature of the real transcriptions have, creating a dataset with diverse forms of speech which could have similar meaning, exposing the classifier to these variations common to the domain context.

After using this method to generate an expanded corpus of artificial calls, we applied Easy Data Augmentation (EDA) to introduce more variability in the dataset and further increase the dataset size [19]. Our implementation of EDA [20] used Brazilian Portuguese synonyms, and was applied three times to triple the final size of the corpus. This approach was used to create a training corpus for the NLU classification models.

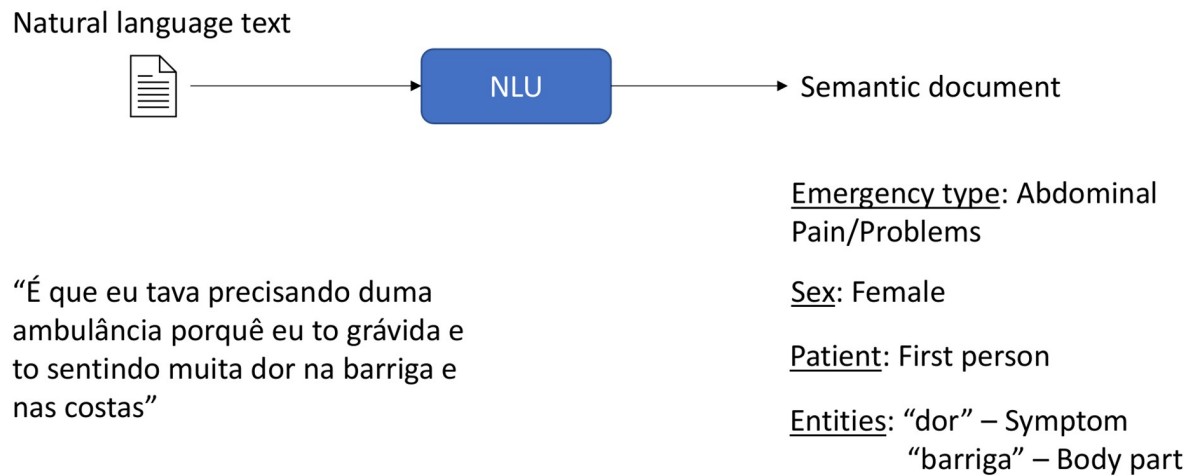

**Fig 3. Sample NLU application for emergency domain.**

### NLU model for Brazilian Portuguese emergency calls

NLU models apply natural language processing algorithms to extract semantic information from text [21]. We selected an intent classification model which aims to predict a classification label for an utterance from a finite set of intent labels [22]. In the emergency call domain, our model aimed to label critical information contained within an emergency call (e.g., gender, symptoms, body parts, timing) (Fig 3).

### Ethics statement

This study was approved by the Research Ethics Committee of the State University of Maringá (UEM) under CAEE: 40184020.0.0000.0104 and all procedures were conducted in accordance with ethical standards. Data provided by SAMU were obtained under a Non-disclosure Agreement signed by all researchers and students involved in the study to protect the confidentiality of the data and the rights of the patients.

## Results

### Manual classification

A total of 182,273 audio files were included in analysis, with a mean duration of 157.60 seconds (SD = 123.31), with a range of 0 to 2,076 seconds. A random subset of 10,010 of these calls were manually classified and 2,326 were classified as emergency calls (Table 1).

### ASM transcription of audio calls in Portuguese

From a total of 2,326 calls, 1926 calls (3.09 hours of audio) were split to our training dataset. Another set of 300 calls (0.60 hours) were used for our evaluation dataset and 100 calls (0.19 hours) were used for testing. The transcription of these audios focused on the main part of the complaint. The Word error rate (WER) was calculated for three different models (Table 2).

### Data augmentation

From the 1,926 calls some of selected emergency types in order to create the BNF grammar. Applying the generation process with this grammar we created a 40,000 artificial example

**Table 1. Initial classifications of manually tanscribed calls to 1-9-2.**

| Classification | % (n) |
| --- | --- |
| Emergency calls | 23.24 (2,326) |
| Sick Person | 7.23 (724) |
| Traffic / Transportation Incidents | 2.32 (232) |
| Falls | 1.92 (192) |
| Psychiatric / Suicide Attemt | 1.67 (167) |
| Convulsions / Seizures | 1.38 (138) |
| Unconscious / Fainting (Near) | 1.37 (137) |
| Overdose / Poisoning (Igestion) | 1.35 (135) |
| Breathing Problems | 0.89 (89) |
| Traumatic Injuries | 0.87 (87) |
| Chest Pain | 0.63 (63) |
| Unknown Problem (Collapse 3rd Party) | 0.55 (55) |
| Hemorrhage / Lacerations | 0.54 (54) |
| Pregnancy / CHildbirth / Miscarriage | 0.50 (50) |
| Assual / Sexual Assual / Stun Gun | 0.47 (47) |
| Abdominal Pain/Problems | 0.39 (39) |
| Stroke (CVA) / Transient Ischemic Attack (TIA) | 0.36 (36) |
| Diabetic Problems | 0.25 (25) |
| Cardiac or Respiratory Arrest / Death | 0.24 (24) |
| Choking | 0.15 (15) |
| Animal Bites / Attacks | 0.11 (11) |
| Allergic Reactions / Envenomations / Stings | 0.09 (9) |
| Burns / Explosions | 0.09 (9) |
| Non-Emergency Calls | 76.76 (7,684) |
| Internal calls | 32.50 (3,253) |
| Others (non-medical calls, non-emergent medical question) | 15.75 (1,577) |
| Mute/Silent Call | 14.01 (1,402) |
| Ambulance delay | 8.00 (801) |
| Duplicate | 2.44 (244) |
| Prank calls | 2.30 (230) |
| Cancellation | 1.77 (177) |

training corpus. This corpus was then enhanced using EDA into 120,000 total emergency calls for training the NLU mode. The same generation and expansion process was used to create a 30,000 test set.

## NLU classification model

The final NLU model was trained on four four selected emergency call categories, to simplify the scope of the problem these were the categories that had the strongest defining

**Table 2. Automatic Speech Recognition results of the testing dataset of emergency calls.**

| Model | Word Error Rate (WER) |
| --- | --- |
| Wav2Vec 2.0 | 42.12% |
| HuBert | 59.96% |
| WavLM | 55.18% |

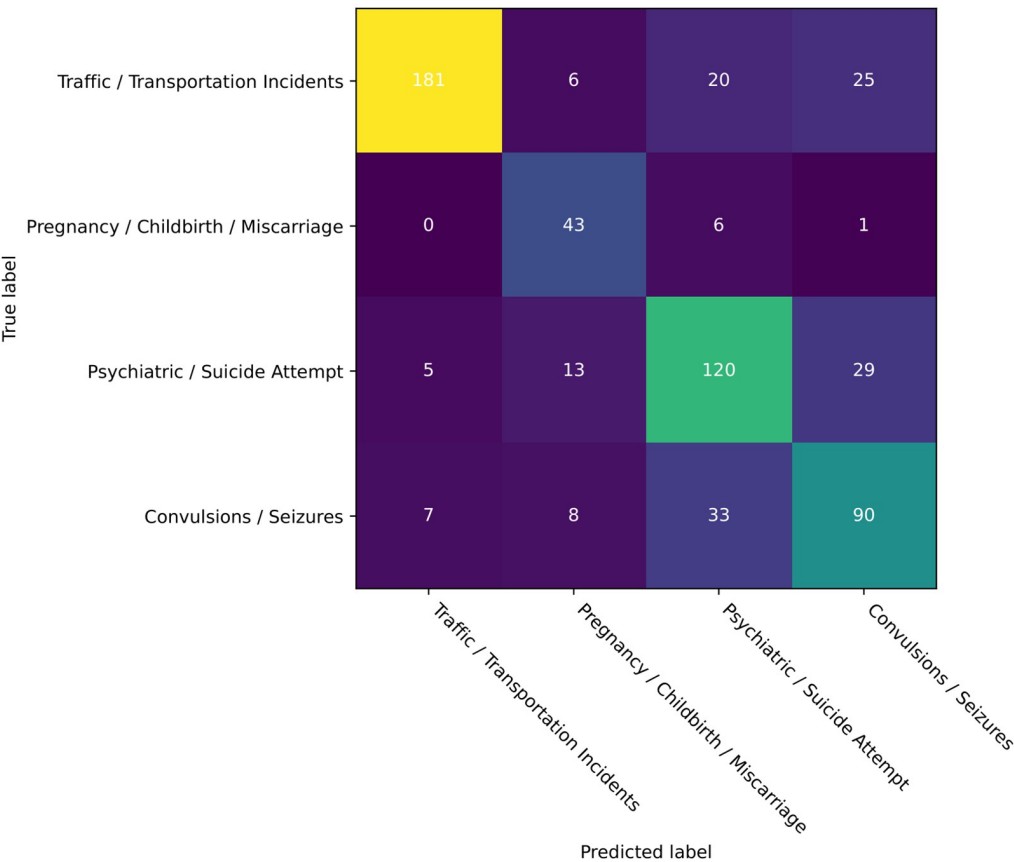

**Fig 4. Confusion matrix for the NLU emergency type classification model validation.**

characteristics and thus easier for manual classification and analysis. Fig 4 shows the results of the classification model for the four selected emergency call categories as a confusion matrix. For validation of the model, we used 232 sentences for the "Traffic / Transportation Incidents" category, 167 for "Psychiatric / Suicide Attempt", 138 for "Convulsions / Seizures", and 50 for "Pregnancy / Childbirth / Miscarriage", for a total of 587 sentences on the test set. In this subset of sentences, the model achieved 73.9% accuracy, 76.0% precision, 73.9% recall, and 74.5% F1 score.

Fig 5 shows the AUCs for each of the four categories assessed with "Psychiatric/Suicide Attempts" and "Traffic/Transportation Incidents" having the highest AUCs.

## Discussion

Our findings provide insight into existing emergency call data in Brazil and demonstrate the feasibility of training a machine learning model to classify emergencies using Brazilian Portuguese language audio recordings. While our findings are focused more on methods and processes, we did obtain interesting data insights. Emergency care in LMICs generally, and the SAMU system in Brazil specifically, remain largely understudied, and there is value in helping to describe this system. First, it was noteworthy that despite the relatively large number of audio recordings available, less than 25% of them were actual emergency calls. This information will be valuable to ensure future work will be adequately powered. Secondly, we found

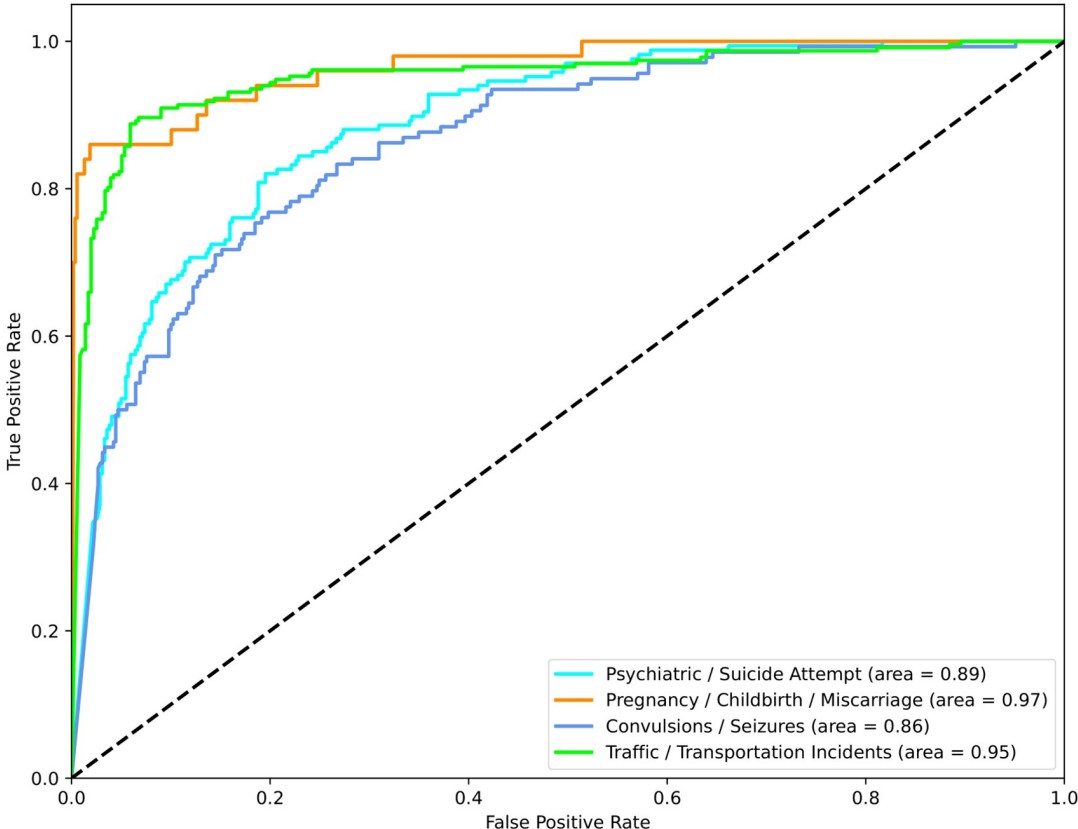

**Fig 5. ROC curves for the NLU emergency type classification.**

that the most common MPDS code was "sick person" at 7% of the total emergency calls. This finding is consistent with previous studies conducted in the United States, where the 'sick person' code was also found to be predominant [23]. This particular code presents certain challenges due to its ambiguous nature, as it could represent a wide range of clinical conditions. While all real-world systems will require some way to code emergencies that are not clearly definable, addressing this category is an analytic challenge for both high- and low-income settings.

In comparison to the situation in Brazil, existing LMICs grapple with similar challenges concerning emergency care. For instance, emergency care in various LMICs has been shown to experience high patient loads and an urgent need for treatment, underscoring the importance of robust and efficient emergency care systems [24]. Additionally, public health emergencies, such as the 2018 Ebola epidemic in Equateur Province in the Democratic Republic of the Congo, highlight the challenges faced by LMICs, where data quality and integrated data management systems have been identified as concerns [25]. These findings not only emphasize the significance of our study within the Brazilian context but also underscore the need for adaptable and scalable solutions for LMICs at large.

In this study we describe a workable, basic framework that adapts existing open-source, high-performing, English language-trained ASR and NLU models. Using this approach we were able to transform unstructured Brazilian Portuguese audio calls into structured, labeled emergency call categories. While our methods are not yet completely optimized, they represent

novel work in this domain and can serve as a foundation for future efforts. We found that the Wave2vec model produced the lowest WER compared to two other ASR models. We also found that accepted data augmentation methods (EDA) enabled us to produce a large corpus of artificial sentences. The final NLU classification model for classification proved to be accurate with the AUCs of all four emergency call categories ranging from 0.86 to 0.97. These findings lay the groundwork for developing a real-time audio classification model for emergency calls in Brazilian Portuguese.

It is imperative to acknowledge that the deployment of our methodology doesn't negate the inherent need for professional validation during emergency calls. Instead, the proposed system aims to act as a preliminary call classification facilitator, enhancing response agility. An accuracy rate of 73.9% indicates notable efficacy, but it also points to the potential for errors in certain instances. Therefore, ongoing human oversight or validation is essential to counteract potential missteps, whether in the transcription phase or the subsequent classification. This dual approach, blending AI capabilities with human discernment, proves especially valuable in challenging scenarios [26]. For instance, in calls originating from noisy environments or with poor signal quality, the model can offer crucial insights to the healthcare professional, assisting them in navigating through uncertainty. Similarly, in situations of truncated or failed communication, the system's imprecise or absent classification can serve as an indicator for the responder to gather additional information from the caller. Thus, rather than replacing human judgment, our proposal seeks to complement it, potentially enhancing healthcare professionals' efficiency and accuracy in critical situations.

The computational models developed within the scope of this study were cost-effective, utilizing only open-source resources that are freely available, which augments their feasibility for deployment in resource-constrained environments. The scarcity of non-English options in freely available voice recognition software presents significant challenges for many LMICs. This framework, initially developed for Portuguese, could be adapted and translated to other settings, including less common languages or dialects, thus overcoming many of the cost and expertise barriers that remain challenging in these settings [27].

## Limitations

Equally notable as our successes were the challenges faced. There exists no universal standard for EMS dispatch coding categories, nor any universal standard for how this data should or could be mapped to hospital-based coding systems. There is no generally accepted lexicon of words used in EMS (in English or other languages) for use in data augmentation. Ultimately, these challenges combined to make our candidate methods rely heavily on artificial sentences built out of our own data, which likely biases our results. Addressing these challenges through collaborative efforts, data sharing and external validation are all next steps for future work in AI processing of EMS calls across languages and resource settings. While our research assistants did not receive any formal training in MPDS, the categories were double-checked after the NLU model classification to ensure that the original manual classification was accurate. This was done by both listening to the call again and by re-reading the transcript.

Our current study focused on four very distinct chief complaints as a proof of concept. These complaints were selected because they were all relatively common, but also conceptually distinct. With an approach of a proof of concept, we elected to work with categories that were most likely to be successful in analysis. With our next steps, we plan to continue to collect audio data to refine our ASR model. This will allow us to create a larger corpus of transcribed and artificial sentences and thus add additional chief complaints. Finally, our ability to manually transcribe audio calls limited our overall dataset of true emergency calls to just over 2,300.

Improved audio quality and increased total audio time will likely improve model performance in the future.

## Conclusions

Through adaptation of open-sourced, freely available English language ASR and NLU models, we were able to transcribe and classify a subset of Portuguese language emergency calls to a ERC in Paraná, Brazil. Our NLU model was highly accurate in differentiating between four disparate chief complaints as an initial use case for the model. We were also able to create a large corpus of artificial sentences to train the NLU model in order to augment a limited dataset of transcribed calls. This framework could also be adapted and applied to other settings in which transcription of large numbers of audio calls may not be possible. Overall, our findings lay the groundwork towards further training of the model and eventual real-time transcription and classification deployment to augment human decision-making. Future work should evaluate accuracy of the model by comparing our automated classifications with ambulance primary impressions and emergency department diagnoses. Also, study the applicability of other machine learning models for NLU like ChatGPT and for ASR like OpenAI Whisper.

## Acknowledgments

The authors acknowledge the SAMU of Maringá, Brazil for providing data used in this study; Duke University's Compute Cluster for their assistance in data processing; and funding from the Duke Global Health Institute AI Pilot grant. They would also like to thank the research students from the State University of Maringá (UEM) who contributed to this study.

## Author Contributions

**Conceptualization:** Dalton Breno Costa, Felipe Coelho de Abreu Pinna, Anjni Patel Joiner, Brian Rice, João Ricardo Nickenig Vissoci, João Carlos Néto.

**Data curation:** Dalton Breno Costa, Felipe Coelho de Abreu Pinna, João Vítor Perez de Souza, Júlia Loverde Gabella, João Ricardo Nickenig Vissoci, João Carlos Néto.

**Formal analysis:** Dalton Breno Costa, Felipe Coelho de Abreu Pinna, João Carlos Néto.

**Funding acquisition:** Anjni Patel Joiner, Luciano Andrade, João Ricardo Nickenig Vissoci.

**Investigation:** Dalton Breno Costa, Felipe Coelho de Abreu Pinna, Anjni Patel Joiner, Brian Rice, João Vítor Perez de Souza, João Carlos Néto.

**Methodology:** Dalton Breno Costa, Felipe Coelho de Abreu Pinna, João Ricardo Nickenig Vissoci, João Carlos Néto.

**Project administration:** Dalton Breno Costa, Felipe Coelho de Abreu Pinna, Luciano Andrade, João Ricardo Nickenig Vissoci, João Carlos Néto.

**Resources:** Luciano Andrade, João Ricardo Nickenig Vissoci.

**Supervision:** Anjni Patel Joiner, Brian Rice, João Ricardo Nickenig Vissoci, João Carlos Néto.

**Validation:** João Ricardo Nickenig Vissoci.

**Visualization:** Dalton Breno Costa, Felipe Coelho de Abreu Pinna.

**Writing – original draft:** Dalton Breno Costa, Felipe Coelho de Abreu Pinna, Anjni Patel Joiner, Brian Rice, João Vítor Perez de Souza, Júlia Loverde Gabella, João Ricardo Nickenig Vissoci, João Carlos Néto.

**Writing – review & editing:** Dalton Breno Costa, Felipe Coelho de Abreu Pinna, Anjni Patel Joiner, Brian Rice, João Vítor Perez de Souza, Júlia Loverde Gabella, Luciano Andrade, João Ricardo Nickenig Vissoci, João Carlos Néto.

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
