## [Decision Letter · Decision Letter 0]

8 Aug 2023

PDIG-D-23-00270

AI-based approach for transcribing and classifying unstructured emergency call data: A methodological proposal

PLOS Digital Health

Dear Dr. Vissoci,

Thank you for submitting your manuscript to PLOS Digital Health. After careful consideration, we feel that it has merit but does not fully meet PLOS Digital Health's publication criteria as it currently stands. Therefore, we invite you to submit a revised version of the manuscript that addresses the points raised during the review process.

Please submit your revised manuscript within 30 days Sep 07 2023 11:59PM. If you will need more time than this to complete your revisions, please reply to this message or contact the journal office at digitalhealth@plos.org. Please include the following items when submitting your revised manuscript:

We look forward to receiving your revised manuscript.

Kind regards,

Mengyu Wang, Ph.D.

Academic Editor

PLOS Digital Health

Journal Requirements:

3. We ask that a manuscript source file is provided at Revision. Please upload your manuscript file as a .doc, .docx, .rtf or .tex

Additional Editor Comments (if provided):

The authors need to discuss about at what level of classification accuracy the NLP model can be used in practical situation to replace nurse on the emergency call. And also, the authors need to tone down the conclusion as 88.7% classification accuracy may not represent the different consequences of misclassified categories.

Reviewers' comments:

Reviewer's Responses to Questions

**Comments to the Author**

1. Does this manuscript meet PLOS Digital Health’s publication criteria? Is the manuscript technically sound, and do the data support the conclusions? The manuscript must describe methodologically and ethically rigorous research with conclusions that are appropriately drawn based on the data presented.

Reviewer #1: Yes

2. Has the statistical analysis been performed appropriately and rigorously?

Reviewer #1: Yes

3. Have the authors made all data underlying the findings in their manuscript fully available (please refer to the Data Availability Statement at the start of the manuscript PDF file)?

Reviewer #1: No

4. Is the manuscript presented in an intelligible fashion and written in standard English?

Reviewer #1: Yes

5. Review Comments to the Author

Reviewer #1: The manuscript titled “AI-based approach for transcribing and classifying unstructured emergency call data: A methodological proposal” is original and attempts to leverage contributions that AI and ML is making in language translation and data synthesis to support processing of emergency calls. To improve interest of wider community of health informatics researchers and clinicians in the rest of LMICs, I recommend that the authors include a distinctive discussion on what is happening in other LMICs and draw similarity/difference to state of practice in Brazil. This will support generalizability of the study findings.

The methodological processes presented is rigorous enough to for us to agree that the study findings are scientifically sound. However, we cannot be sure on the appropriateness of the data set created through artificial training corpus building. How did the authors address issues of written text viz a vie common practice of using informal expressions such as slang words in spoken language as would be the case in emergency situations? 

The results show an improvement in the word error rate for their proposed Ai-based model over the other models. However, the discussions did not reflect this as still being high to raise risks to patient life in case such an error in classification happened in highly critical emergency case. It would be appropriate to reflect on this in the writeup. 

The conclusions are based on the finding of the study. And actually, the evidence presented in this work is substantial to warrant such conclusions. however, the lack of availability of the data set and or information about how it can be ethically accessed limits other researchers who might be interested in further exploring the domain.

6. PLOS authors have the option to publish the peer review history of their article (what does this mean?). If published, this will include your full peer review and any attached files.

**Do you want your identity to be public for this peer review?** For information about this choice, including consent withdrawal, please see our Privacy Policy.

Reviewer #1: No

---

## [Editor Report · Decision Letter 1]

7 Nov 2023

AI-based approach for transcribing and classifying unstructured emergency call data: A methodological proposal

PDIG-D-23-00270R1

Dear Dr. Vissoci,

We are pleased to inform you that your manuscript 'AI-based approach for transcribing and classifying unstructured emergency call data: A methodological proposal' has been provisionally accepted for publication in PLOS Digital Health.

Best regards,

Mengyu Wang, Ph.D.

Academic Editor

PLOS Digital Health

The revision is sufficient to address the comments from reviewers.